# The coronavirus disease 2019 (COVID-19) vaccination psychological antecedent assessment using the Arabic 5c validated tool: An online survey in 13 Arab countries

Marwa Shawky Abdou[1]*, Khalid A. Kheirallah[2], Maged Ossama Aly[3], Ahmed Ramadan[4], Yasir Ahmed Mohammed Elhadi[5], Iffat Elbarazi[6], Ehsan Akram Deghidy[7], Haider M. El Saeh[8], Karem Mohamed Salem[9], Ramy Mohamed Ghazy[10]

1 Epidemiology Department, High Institute of Public Health, Alexandria University, Alexandria, Egypt, 2 Department of Public Health, Medical School of Jordan University of Science and Technology, Irbid, Jordan, 3 Nutrition Department, High Institute of Public Health, Alexandria University, Alexandria, Egypt, 4 Department of Applied Statistics, Faculty of Graduate Studies for Statistical Research, Cairo University, Giza, Egypt, 5 Department of Public Health, Medical Research Office, Sudanese Medical Research Association, Khartoum, Sudan, 6 Institute of Public Health, College of Medicine and Health Sciences, United Arab Emirates University, AlAin, UAE, 7 Department of Biomedical Informatics and Medical Statistics, Medical Research Institute, Alexandria University, Alexandria, Egypt, 8 Community Medicine Department, Faculty of Medicine, University of Tripoli, Tripoli, Libya, 9 Department of Internal Medicine, Faculty of Medicine, Fayoum University, Faiyum, Egypt, 10 Tropical Health Department, High Institute of Public Health, Alexandria University, Alexandria, Egypt

☯ These authors contributed equally to this work.
* marwa.shawky@alexu.edu.eg, marwa.shawky21@gmail.com

**Data Availability Statement:** All relevant data are within the paper and its Supporting Information files.

## Abstract

### Background

Following the emergency approval of the coronavirus disease 2019 (COVID-19) vaccines, research into its vaccination hesitancy saw a substantial increase. However, the psychological behaviors associated with this hesitancy are still not completely understood. This study assessed the psychological antecedents associated with COVID-19 vaccination in the Arab population.

### Methodology

The validated Arabic version of the 5C questionnaire was distributed online across various social media platforms in Arabic-speaking countries. The questionnaire had three sections, namely, socio-demographics, COVID-19 related infection and vaccination, and the 5C scale of vaccine psychological antecedents of confidence, complacency, constraints, calculation, and collective responsibility.

### Results

In total, 4,474 participants with a mean age of 32.48 ± 10.76 from 13 Arab countries made up the final sample, 40.8% of whom were male. Around 26.7% of the participants were

**Funding:** The author(s) received no specific funding for this work.

**Competing interests:** The authors have declared that no competing interests exist.

found to be confident about the COVID-19 vaccination, 10.7% indicated complacency, 96.5% indicated they had no constraints, 48.8% had a preference for calculation and 40.4% indicated they had collective responsibility. The 5C antecedents varied across the studied countries with the confidence and collective responsibility being the highest in the United Arab Emirates (59.0% and 58.0%, respectively), complacency and constraints in Morocco (21.0% and 7.0%, respectively) and calculation in Sudan (60.0%). The regression analyses revealed that sex, age, educational degrees, being a health care professional, history of COVID-19 infection and having a relative infected or died from COVID-19 significantly predicted the 5C psychological antecedents by different degrees.

## Conclusion

There are wide psychological antecedent variations between Arab countries, and different determinants can have a profound effect on the COVID-19 vaccine's psychological antecedents.

## Introduction

Severe Acute Respiratory Syndrome Coronavirus 2 (SARS-CoV-2) which was initially identified in China in December 2019 is the causative agent of coronavirus disease 2019 (COVID-19). It has affected most countries in the world [1]. Most infected patients usually suffer from mild to moderate flu-like symptoms, including fever, cough, sore throat, anosmia, and ageusia [2, 3]. The global threat of the pandemic is still on the rise with more than 245 million cases and 4.9 million deaths [4]. At more than 16 million cases and nearly 298 thousand deaths, the World Health Organization (WHO) situational report ranked the Eastern Mediterranean Region fourth in the number of COVID-19 cases [5–7]. As of October 9, 2021, in the Arab world, Iraq had the highest number of cases followed by Morocco, Jordan, the United Arab Emirates (UAE), and Tunisia, with COVID-19 associated deaths being highest in Tunisia, Iraq, Egypt and Morocco (25,028, 22,537, 17,658 and 14,443, respectively) [7]. COVID-19 has mutated over time into new variants, which to date have been recognized as the alpha, beta, gamma, and delta variants [8]. As each new variant has posed a further threat on a global level, immediate precautionary measures such as vaccination have been put in place in all countries [9].

As the non-pharmaceutical intervention measures against COVID-19, such as social distancing and curfews, have not been enough to mitigate the virus spread [10], there is a global consensus that COVID-19 vaccination is the most effective approach to control the pandemic [11]. The unprecedented research efforts and global coordination have resulted in the rapid development and administration of vaccines to control COVID-19 [12]. Since its emergence, COVID-19 has resulted in a surge in vaccine developments, with many undergoing pre-clinical developments, of which 43 have entered clinical trials, including some approaches that have not previously been licensed for humans [13].

These focused scientific efforts have given rise to different vaccine modalities and novel techniques. The approved COVID-19 vaccines are either mRNA (manufactured by Moderna and BioNTech/Pfizer), inactivated viruses (Sinovac, Sinopharm), viral vectors (Oxford/Astra-Zeneca, Gamaleya, Janssen/Johnson & Johnson, CanSino), or protein sub-units (Novavax). The vaccine produced by BioNTech/Pfizer was the first licensed COVID-19 vaccine deployed

to the public [14], and since then many more countries, such as Cuba and Brazil, have entered the vaccine development race [15]. The efficacy of vaccines varied between different types from more than 70% to more than 90%. Moreover, all of them had a variable protection against mutated variants [16]. However, as there is no gold standard for titration of IgG serum antibodies or T-cell response, it seems to be difficult to compare the immune responses of the different vaccines [17]. Another note that efficacy of vaccines varies according to the studied population, defining outcomes, and design which share in explaining the variation in the efficacy of vaccines [6]. The production of effective vaccines is useless in case of being unaffordable to individuals around the globe which put a huge burden on governments to ensure delivering the vaccines to their population [18].

People all over the world have expressed concerns about the authorized COVID-19 vaccines for several reasons [13], such as the rapid development and release of the vaccines and conspiracy theories about their origin. Vaccine hesitancy (VH), which existed before the COVID-19 emergence, is the term applied to people who doubt the veracity of vaccines and stop themselves —and influence others—from getting vaccinated [19]. One of the major obstacles to successful vaccination programs has been VH, as these people are at a greater risk of being seriously infected and spreading the virus into the community [20]. As such, VH was nominated as one of the nine global public health threats by the WHO in 2019 [21] and is defined as the behavior associated with a delay in accepting or a refusal to vaccinate despite available services. It is a complex and context-specific behavior that varies across time, place, and disease but is still influenced by factors such as complacency, convenience, and confidence [22].

The vaccination tool (5C model) developed by Betsch et al. [23] identifies five psychological antecedents that influence a person's choice to be vaccinated or not; confidence, complacency, constraints, calculation, and collective responsibility. The 5C scale has been used to provide insights into how people think, feel, and behave toward vaccination. These antecedents have been found to impact vaccination behavior to varying degrees and reveal the mental portrayals, attitudinal and behavioral propensities from the environments and contexts they live in [23–25]. These antecedents are now being widely used as the framework to assess VH in high-income countries to determine the possible COVID-19 vaccine take up rates [26]. Many studies had used the 5C scale to assess COVID-19 vaccine psychological antecedants [27–29], however, there is no any studies conducted among arab population using the validated arabic version of the 5C scale.

The COVID-19 VH rate has been found to differ significantly due to sociodemographic characteristics, seasonal flu vaccination statuses, COVID-19 risk perceptions, and the perceived benefits of and clinical barriers to the COVID-19 vaccine [30]. In Hong Kong, 63% of nursing staff claimed that they were likely to take the COVID-19 vaccine when it was available [31], and in many low-and middle-income countries, the vaccine acceptance rate has been found to range from 66.5% in Burkina Faso to 96.6% in Nepal, with an overall acceptance rate of 80.3% [32]. At this stage in the pandemic, especially as vaccine compliance remains variable and inconsistent, public health officers and policymakers, especially in developing countries where healthcare resources are limited, need to understand the reasons and factors associated with VH. This study was therefore developed to investigate the psychological antecedent factors in Arab populations toward the COVID-19 vaccination.

## Materials and methods

### Study design, sampling, and data collection

A cross-sectional, web-based (through Qualtrics), anonymous survey using the Arabic-validated version of the 5C questionnaire [33] was conducted between December 2020 and

February 2021. The survey was distributed via email and social media, Facebook, Twitter, and WhatsApp between December 2020 and February 2021. The sample size was calculated using EpiInfo version 7.2. The minimum required sample size was 700 based on the following criteria: population size of 440 million, predicted frequency of 35.0% [34], design effect of 2, confidence level of 95%, and margin of error of 5%. To adjust for any stratification and to eliminate any invalid replies, we increased the sample size several times. Subjects aged 18 years old or above and residing in Arab countries during the COVID-19 pandemic were eligible for participation. Study participants were allowed to fill the questionnaire after reading and consenting to the online informed consent. There was no compensation for taking part in this research, and it was it is not permitted to complete more than one survey.

## Data collection tools

The survey comprised the following sections: sociodemographic characteristics (age, sex, residence, level of education, marital status, occupation, and presence of comorbidities); past COVID-19 infection and vaccination history (previous infection, family history, mortality, influenza vaccination, types of COVID-19 vaccines, searching web for information about COVID-19 vaccine); and 15 questions covering the five 5C domains; confidence, complacency, constraints, calculation, and collective responsibility; each of which had three questions to be answered using a 7-point Likert scale (1; strongly disagree to 7; strongly agree). The cut-off point for the confidence, complacency, constraints, calculation, and collective responsibility domains were 5.7, 4.7, 6.0, 6.3, and 6.2, respectively [35].

## Definitions

1. **Confidence:** It refers to trust in the vaccine, reliability, and effectiveness [22] as well as trust in health care system and health care professionals. Lack of trust and mistrust leads to lower uptake of the vaccine and lower confidence in the health care system and more acceptance of misinformation.

2. **Constraint:** It refers to structural and psychological barriers that may hinder people getting vaccinated even if they have the intention to [36]. Such barriers may include access, time, self-efficacy, empowerment and lack of behavioral control.

3. **Complacency:** as defined by Betsch "perceived risks of vaccine-preventable diseases are low and vaccination is not deemed a necessary preventive action" in other words disease is perceived as low risk impacting on vaccination uptake as the person may considered it to be not necessary.

4. **Calculation:** It implies that people search for information to compare the risk of infections versus vaccination to make an informed decision [23] It is argued that calculation can be a sign of risk averse individuals and may have a negative impact on vaccination behavior.

5. **Collective responsibility:** defined by Betsch et al as "the willingness to protect others by one's own vaccination by means of herd immunity. The flipside is the willingness to have a free ride when a sufficient number of other people are vaccinated" in other words: It refers to people who vaccinate themselves intending to protect others and appreciating the role of herd immunity and limiting transmission [23].

## Statistical analysis

The collected data was wrangled, coded, and analyzed using the Python 3.9.2 software. The quantitative variables, expressed using mean ± SD whereas counts (%), were utilized to describe the categorical variables; a chi-square test was used to estimate the pairwise correlations between the categorical variables; and the respondents were categorized (Yes/No) based on their mean 5C scores with reference to the cutoff points. Five stepwise binary logistic regression models using all the variables were conducted to estimate the significant predictors for confidence, complacency, calculation, constraints, and collective responsibility, odds ratios and 95% confidence intervals (OR, 95% CI) were reported, and $P < 0.05$ was considered statistically significant.

## Ethical considerations

This study was approved by the Ethics Committee of the Faculty of Medicine, Alexandria University, Egypt (IRB No: 00012098). The study was performed in accordance with the ethical standards laid down in the 1964 Declaration of Helsinki and its later amendments or comparable ethical standards [37]. All participants were informed that their participation was voluntary, and consent was obtained by answering a question prior to administering the survey. If participants accepted to participate in the survey, the survey link was provided, and a refusal to participate terminated access to the online questionnaire.

# Results

## Respondent characteristics

A total of 4,474 participants with a mean age of 32.48 ± 10.76 years from 13 Arabic countries were included in the current analysis, of which 40.8% were males and 7.6% (330) participants received at least one dose of COVID-19 vaccines. The majority were married (50.7%), half held a university degree (50.3%), reported no chronic illnesses (82.7%), were health care professionals (HCPs; 40.0%), reported being vaccinated against the COVID-19 infection (27.9%), had at least one relative who had been infected with COVID-19 (47.5%), had at least one relative who had died from COVID-19 (33.6%), and knew that there were different types of COVID-19 vaccines (21.5%) (Table 1).

## Psychological vaccination antecedents

Fig 1 indicates that 1.197 (26.7%) participants were confident about the COVID-19 vaccination, 477 (10.7%) were complacent, 4319 (96.5%) had no constraints, 2185 (48.8%) indulged in calculation, and 1810 (40.4%) had collective responsibility.

## Psychological antecedents to COVID-19 vaccination among the studied countries

As shown in Fig 2, confidence and collective responsibility were higher in the UAE (59% and 58%), complacency and constraint in Morocco (21% and 7%), and calculation in Sudan (60%). Egypt had the lowest confidence (15%), Lebanon had the lowest complacency (7.5%), Sudan had the lowest constraint (1.2%), Iraq had the lowest calculation (36%), and Morocco had the lowest collective responsibility (25%).

**Table 1. Participant background characteristics.**

| Variables (n = 4,474)* | Number | % |
|---|---|---|
| | (n = 4,474) | |
| **Sex** | | |
| Male | 1,825 | *40.8* |
| Female | 2,649 | *59.2* |
| **Age (years) Mean ± SD** | 32.48 ± 10.76 | |
| **Marital status** | | |
| Single | 1,804 | *44.4* |
| Married | 2,058 | *50.7* |
| Divorced | 139 | *3.4* |
| Widowed | 60 | *1.5* |
| **Educational status** | | |
| Pre-university | 323 | *7.2* |
| Technical/ vocational education | 129 | *2.9* |
| University degree | 2,243 | *50.3* |
| Postgraduate degree | 1,502 | *33.7* |
| Other | 266 | *6.0* |
| **Have a chronic disease** | 773 | *17.3* |
| **Be a health care professional** | 1,789 | *40.0* |
| **Previously infected with COVID-19** | 1,025 | *27.9* |
| **Relative infected with COVID-19** | 1,915 | *47.5* |
| **Relative died from COVID-19** | 1,390 | *33.6* |
| **Received the influenza vaccine** | 104 | *26.1* |
| **Know the different COVID-19 vaccines** | 3510 | *78.5* |
| The best COVID-19 vaccine (n = 330)* | | |
| Moderna | 24 | *7.3* |
| Pfizer- BioNTech | 194 | *58.8* |
| Oxford-AstraZeneca | 52 | *15.8* |
| Sinopharm | 51 | *15.5* |
| Sputnik V | 9 | *2.7* |
| **Infected with COVID-19 after vaccination** | 71 | *21.5* |
| **Know the COVID-19 vaccine instructions** | 184 | *55.8* |
| **Conducted internet search on COVID-19 vaccine** | 199 | *60.3* |
| **Got COVID-19 vaccine as it is free** | 207 | *62.7* |

*Counts and percentages are presented for the actual number of responders to each variable

## Bivariate analysis of the 5C domains and the independent variables

Table 2 shows the distribution for each of the 5C domains by the independent variables at the bi-variate levels.

 **Confidence.** The following variables were found to significantly affect the confidence domain: male gender ($P < 0.001$); marital status ($P = 0.006$); educational level ($P = 0.003$); previous history of COVID-19 infection ($P < 0.001$); relatives died from COVID-19 infection ($P = 0.033$); taking annual influenza vaccine ($P = 0.007$); knowing about the different vaccine types ($P = 0.016$); following COVID-19 protective measures ($P < 0.001$); internet search for COVID-19 related information ($P < 0.001$); and vaccine cost ($P < 0.001$).

 **Complacency.** Complacency was significantly predicted by: educational level ($P = 0.011$); being a HCP ($P < 0.001$); previous history of COVID-19 infection ($P = 0.034$); knowing about

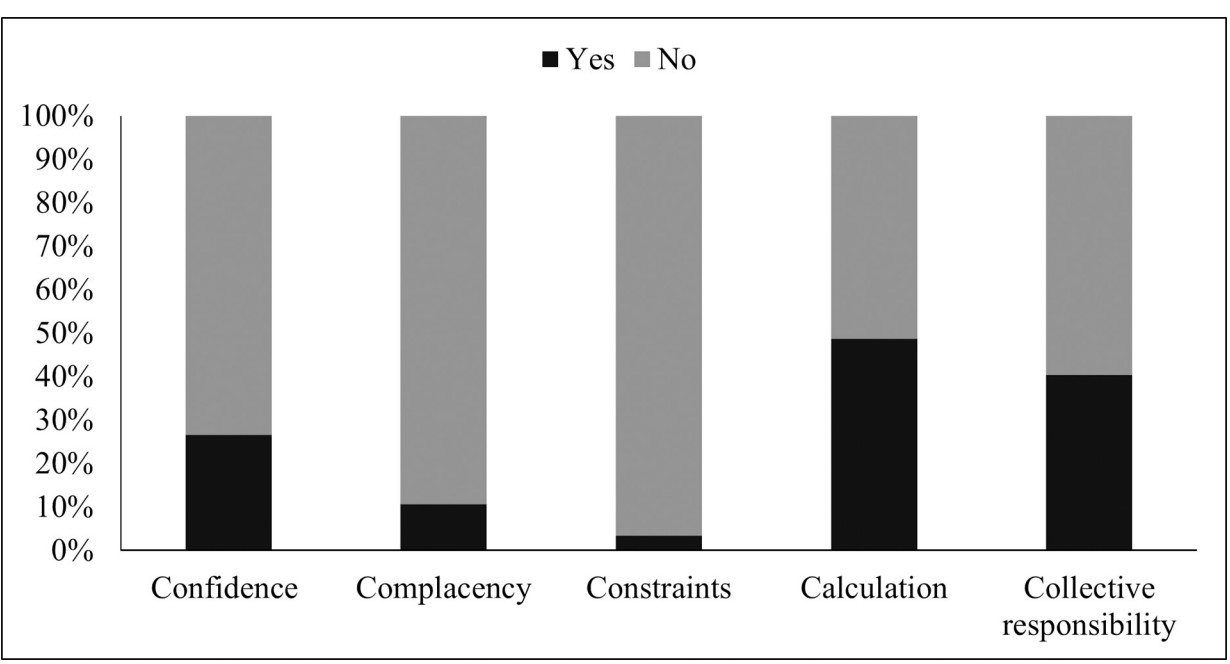

**Fig 1. Psychological antecedents for vaccination.**

the different vaccine types ($P < 0.001$); following COVID-19 protective measures ($P = 0.003$); and internet searches for COVID-19 related information ($P = 0.02$).

**Constraints.** The COVID-19 related constraint was significantly affected by: being HCP ($P = 0.004$); being previously infected with COVID-19 ($P = 0.001$); and knowing about the different vaccine types ($P < 0.001$).

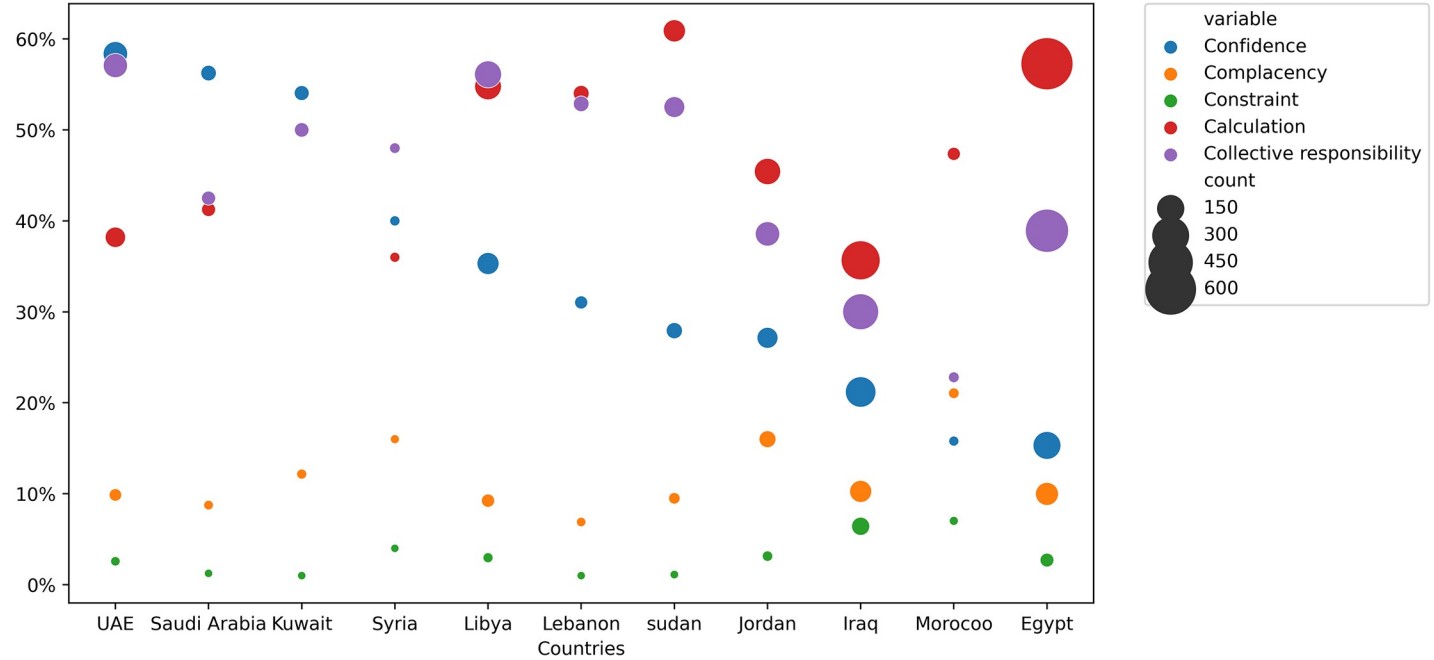

**Fig 2. COVID-19 vaccination psychological antecedents in the studied countries.**

**Table 2. Distribution for each of the 5C domains by the independent variables at the bi-variate levels.**

| Variable | Confidence n (%) | | Complacency n (%) | | Constraints n (%) | | Calculation n (%) | | Responsibility n (%) | |
|---|---|---|---|---|---|---|---|---|---|---|
| | **Yes** | **No** | **Yes** | **No** | **Yes** | **No** | **Yes** | **No** | **Yes** | **No** |
| **Sex** | | | | | | | | | | |
| Male | 560 (30.68) | 1265 (69.32) | 209 (11.45) | 1616 (88.55) | 58 (3.18) | 1767 (96.82) | 817 (44.77) | 1008 (55.23) | 746 (40.88) | 1079 (59.12) |
| Female | 637 (24.05) | 2012 (75.95) | 268 (10.18) | 2381 (89.88) | 98 (3.70) | 2551 (96.30) | 1368 (51.64) | 1281 (48.36) | 1064 (40.17) | 1585 (59.83) |
| *P-value* | <0.001 | | 0.169 | | 0.394 | | 0.007 | | 0.656 | |
| **Marital status** | | | | | | | | | | |
| Single | 453 (25.11) | 1351 (74.89) | 185 (10.25) | 1619 (89.75) | 70 (3.88) | 1734 (96.12) | 832 (45.62) | 972 (54.38) | 733 (40.63) | 1071 (59.37) |
| Married | 545 (26.48) | 1513 (73.52) | 225 (10.93) | 1833 (89.07) | 68 (3.30) | 1990 (96.70) | 1037 (50.39) | 1021 (49.61) | 838 (40.72) | 1220 (59.28) |
| Divorced | 35 (25.18) | 104 (74.82) | 15 (10.79) | 124 (89.21) | 5 (3.60) | 134 (96.40) | 57 (41.01) | 82 (58.99) | 43 (30.94) | 96 (69.06) |
| Widow | 27 (45) | 33 (55) | 4 (6.67) | 56 (93.33) | 1 (1.67) | 59 (98.33) | 33 (55) | 27 (45) | 25 (41.67) | 35 (58.33) |
| *P-value* | 0.006 | | 0.693 | | 0.668 | | 0.011 | | 0.148 | |
| **Educational status** | | | | | | | | | | |
| Pre-university | 108 (33.44) | 215 (66.56) | 46 (14.24) | 277 (85.76) | 9 (2.79) | 314 (97.21) | 138 (42.72) | 185 (57.28) | 129 (39.94) | 194 (60.06) |
| Technical/ vocational education | 45 (34.88) | 84 (65.12) | 16 (12.40) | 113 (87.60) | 5 (3.88) | 124 (96.12) | 50 (38.76) | 79 (61.24) | 53 (41.09) | 76 (58.91) |
| University degree | 581 (25.90) | 1662 (74.10) | 269 (11.99) | 1974 (88.01) | 83 (3.70) | 2160 (96.30) | 1063 (47.39) | 1180 (52.61) | 904 (40.30) | 1339 (59.70) |
| Postgraduate degree | 391 (26.03) | 1111 (73.97) | 119 (7.92) | 1383 (92.08) | 43 (2.86) | 1459 (97.14) | 839 (55.86) | 663 (44.14) | 648 (43.14) | 854 (56.86) |
| Other | 71 (26.69) | 195 (73.31) | 27 (10.15) | 239 (89.85) | 16 (6.02) | 250 (93.98) | 93 (34.96) | 173 (65.04) | 75 (28.20) | 191 (71.80) |
| *P-value* | 0.003 | | 0.011 | | 0.107 | | <0.001 | | <0.001 | |
| **Being a HCP** | 459 (25.66) | 1330 (74.34) | 135 (7.55) | 1654 (92.45) | 45 (2.52) | 1744 (97.48) | 956 (53.44) | 833 (46.56) | 815 (45.56) | 974 (54.44) |
| *P-value* | 0.187 | | <0.001 | | 0.004 | | <0.001 | | <0.001 | |
| **Get COVID** | 198 (19.32) | 827 (80.68) | 126 (12.29) | 899 (87.71) | 55 (5.37) | 970 (94.63) | 454 (44.29) | 571 (55.70) | 357 (34.83) | 668 (65.17) |
| *P-value* | <0.001 | | 0.0341 | | 0.001 | | 0.069 | | <0.001 | |
| **Relative infected with COVID** | 503 (26.27) | 1412 (73.73) | 187 (9.77) | 1728 (90.23) | 70 (3.67) | 1845 (96.34) | 938 (48.98) | 977 (51.02) | 806 (42.09) | 1109 (57.91) |
| *P-value* | 0.06 | | 0.095 | | 0.392 | | 0.361 | | 0.102 | |
| **Relative died from COVID** | 342 (24.60) | 1048 (75.40) | 146 (10.50) | 1244 (89.50) | 52 (3.74) | 1338 (96.26) | 741 (53.31) | 649 (46.69) | 584 (39.42) | 806 (60.58) |
| *P-value* | 0.033 | | 0.937 | | 0.316 | | 0.002 | | 0.395 | |
| **Getting Flu vaccine** | 35 (33.65) | 69 (66.35) | 10 (9.62) | 94 (90.38) | 5 (4.81) | 99 (95.19) | 52 (50) | 52 (50) | 42 (40.38) | 62 (59.62) |
| *P-value* | 0.007 | | 0.587 | | 0.911 | | 0.252 | | 0.049 | |
| **Knowing different COVID vaccines** | 969 (27.61) | 2541 (72.39) | 337 (9.60) | 3173 (90.40) | 96 (2.74) | 3414 (97.26) | 1814 (51.68) | 1696 (48.32) | 1521 (43.33) | 1989 (56.67) |
| *P-value* | 0.016 | | <0.001 | | <0.001 | | <0.001 | | <0.001 | |
| **Best COVID vaccine** | | | | | | | | | | |
| Moderna | 10 (41.67) | 14 (58.33) | 2 (8.33) | 22 (91.67) | 0 (0) | 24 (100) | 10 (41.67) | 14 (58.33) | 13 (54.17) | 11 (45.83) |
| Pfizer- BioNTech | 69 (35.57) | 125 (64.43) | 13 (6.70) | 181 (93.30) | 3 (1.55) | 191 (98.45) | 117 (60.31) | 77 (39.69) | 94 (48.45) | 100 (51.55) |
| Oxford-AstraZeneca | 8 (15.38) | 44 (84.62) | 4 (7.69) | 48 (92.31) | 1 (1.92) | 51 (98.08) | 30 (57.69) | 22 (42.31) | 24 (46.15) | 28 (53.84) |
| Sinopharm | 25 (49.02) | 26 (50.98) | 7 (13.73) | 44 (86.27) | 1 (1.96) | 50 (98.04) | 29 (56.86) | 22 (43.14) | 24 (47.06) | 27 (52.94) |
| Sputnik V | 0 (0) | 9 (100) | 1 (11.11) | 8 (88.89) | 2 (22.22) | 7 (77.78) | 4 (44.44) | 5 (55.56) | 1 (11.11) | 8 (88.89) |

(*Continued*)

**Table 2.** (Continued)

| Variable | Confidence n (%) | | Complacency n (%) | | Constraints n (%) | | Calculation n (%) | | Responsibility n (%) | |
|---|---|---|---|---|---|---|---|---|---|---|
| | Yes | No | Yes | No | Yes | No | Yes | No | Yes | No |
| *P-value* | <0.001 | | 0.596 | | 0.001 | | 0.444 | | 0.256 | |
| **Getting COVID after vaccination** | 17 (35.42) | 31 (64.58) | 6 (12.50) | 42 (87.50) | 1 (2.08) | 47 (97.92) | 15 (31.25) | 33 (68.75) | 20 (41.67) | 28 (58.33) |
| *P-value* | 1 | | 1 | | 0.828 | | 0.008 | | 0.981 | |
| **COVID vaccine Instructions** | 64 (51.20) | 61 (48.80) | 8 (6.40) | 117 (93.60) | 2 (1.60) | 123 (98.40) | 61 (48.80) | 64 (51.20) | 72 (57.60) | 53 (42.40) |
| *P-value* | <0.001 | | 0.003 | | 0.479 | | 0.975 | | <0.001 | |
| **Internet search about COVID vaccine** | 60 (44.44) | 72 (55.65) | 23 (17.04) | 112 (82.96) | 2 (1.48) | 133 (98.52) | 71 (52.59) | 64 (47.41) | 64 (47.41) | 71 (52.59) |
| *P-value* | <0.001 | | 0.02 | | 0.345 | | 0.25 | | 0.119 | |
| **Getting COVID vaccine if free** | 61 (43.57) | 79 (56.43) | 18 (12.86) | 122 (87.14) | 2 (1.43) | 138 (98.57) | 70 (50.00) | 70 (50.00) | 71 (50.71) | 69 (49.29) |
| *P-value* | <0.001 | | 1 | | 0.285 | | 0.835 | | 0.003 | |

**Calculation.** The calculation domain was significantly affected by: gender ($P = 0.007$); marital status ($P = 0.011$); educational level ($P < 0.001$); being a HCP ($P < 0.001$); having at least one relative die due to COVID-19 ($P = 0.002$); knowing about the different available vaccines ($P < 0.001$); and believing that there was a risk of getting COVID-19 even after vaccination ($P = 0.008$).

**Collective responsibility.** The collective responsibility domain was significantly affected by: educational level ($P < 0.001$); working as a HCP ($P < 0.001$); being previously infected with COVID-19 ($P < 0.001$); receiving a yearly influenza vaccine ($P = 0.049$); knowing about the different vaccine types ($P < 0.001$); following COVID-19 protective measures ($P < 0.001$); and the availability of free vaccines ($P = 0.003$).

## Determinants for the psychological vaccination antecedents

Table 3 and S1 Table show the regression analyses for the predictors affecting the psychological antecedents. The significant confidence antecedent predictors were: female gender (OR = 0.701) (95% CI: 0.592–0.829); age (OR = 1.029) (95% CI: 1.019-1.040); university, post-graduate and other education (OR = 0.708, 0.609 and 0.502, respectively) (95% CI: 0.519–0.966, 0.433–0.856 and 0.315–0.800, respectively); previous COVID-19 infection (OR = 0.555)

**Table 3.** 5C domain predictors.

| Independent Variables | Odd Ratio | P-value | 95% C.I. for Odd Ratio |
|---|---|---|---|
| | | Confidence | |
| Constant | 0.065 | <0.001* | |
| **Gender (Female)[a]** | 1.428 | <0.001* | 1.206-1.690 |
| **Age** | 1.029 | <0.001* | 1.019-1.040 |
| **Education[b]** | | 0.024* | |
| Technical/ vocational education | 0.756 | 0.325 | 0.433–1.320 |
| University degree | 0.708 | 0.030* | 0.519–0.966 |
| Postgraduate degree | 0.609 | 0.004* | 0.433–0.856 |
| Others | 0.502 | 0.004* | 0.315–0.800 |
| **Previously infected with COVID-19 [c]** | 0.555 | <0.001* | 0.450–0.686 |
| **Relative infected with COVID-19[d]** | 1.264 | 0.01* | 1.057–1.511 |
| **Relatives died from COVID-19[e]** | 0.803 | 0.019* | 0.669–0.964 |

*(Continued)*

**Table 3.** (Continued)

| Independent Variables | Odd Ratio | P-value | 95% C.I. for Odd Ratio |
|---|---|---|---|
| **Complacency** | | | |
| Constant | 0.144 | <0.001* | |
| **Education[b]** | | 0.045* | |
| Technical/ vocational education | 0.705 | 0.377 | 0.325–1.531 |
| University degree | 0.730 | 0.138 | 0.483–1.106 |
| Postgraduate degree | 0.496 | 0.004* | 0.309–0.799 |
| Others | 0.732 | 0.306 | 0.403–1.329 |
| **HCP[f]** | 0.512 | <0.001* | 0.387-0.678 |
| **previously infected with COVID-19 [c]** | 1.556 | 0.002* | 1.171-2.068 |
| **Constraints** | | | |
| Constant | 0.03 | <0.001* | |
| **HCP[f]** | 0.518 | 0.008* | 0.319–0.841 |
| **Previously infected with COVID-19 [c]** | 2.309 | <0.001* | 1.461–3.649 |
| **Calculation** | | | |
| Constant | 0.348 | <0.001* | |
| **Sex (Female)[a]** | 1.362 | <0.001* | 1.169-1.586 |
| **Age** | 1.012 | 0.014* | 1.002-1.021 |
| **Education[b]** | | <0.001* | |
| Technical/ vocational education | 0.697 | 0.188 | 0.407–1.193 |
| University degree | 1.167 | 0.302 | 0.870–1.565 |
| Postgraduate degree | 1459 | 0.020* | 1.061–2.007 |
| Others | 0.757 | 0.194 | 0.497–1.153 |
| **HCP[f]** | 1.268 | 0.003* | 1.082-1.486 |
| **Relative died from COVID-19[e]** | 1.248 | 0.007* | 1.064-1.463 |
| **Collective responsibility** | | | |
| Constant | 0.439 | <0.001* | |
| **Age** | 1.014 | 0.005* | 1.004-1.023 |
| **HCP[f]** | 1.594 | <0.001* | 1.358-1.872 |
| **Previously infected with COVID-19[c]** | 0.613 | <0.001* | 0.510-0.736 |
| **Relative infected with COVID-19[d]** | 1.261 | 0.005* | 1.072-1.482 |

*Statistically significant

[a]ref; Male

[b]ref; Pre-university education

[c]ref; Not getting COVID

[d]ref: relative not getting COVID

[e]ref; no relative died

[f]ref; No HCP.

(95% CI: 0.450-0.686); having a relative infected with COVID-19 (OR = 1.264) (95% CI 1.057-1.511); and having a relative die from COVID-19 (OR = 0.803) (95% CI 0.669–0.964). The significant complacency predictors were: having a postgraduate degree (OR = 0.496); (95% CI 0.309-0.799); being a HCP (OR = 0.512) (95% CI: 0.387-0.678); and being previously infected with COVID-19 (OR = 1.556) (95% CI: 1.171-2.068). The significant constraint predictors were: being a HCP (OR = 0.518) (95% CI 0.319-0.841); and having been infected with COVID-19 (OR = 2.309) (95% CI 1.461-3.649). The significant calculation predictors were; being female (OR = 1.362) (95% CI 1.169-1.586); age (OR = 1.012) (95% CI 1.002-1.021);

having a postgraduate degree (OR = 1.459) (95% CI 1.061-2.007); being a HCP (OR = 1.268) (95% CI 1.082-1.486); and having a relative die from COVID-19 (OR = 1.248) (95% CI 1.064-1.463). The significant collective responsibility predictors were: age (OR = 1.014) (95% CI 1.004-1.023); being a HCP (OR = 1.594) (95% CI 1.358-1.872); having been infected with COVID-19 (OR = 0.613) (95% CI 0.510-0.736); and having a relative infected with COVID-19 (OR = 1.261) (95% CI 1.072-1.482).

## Discussion

Tools such as the 5C can determine the psychological antecedents toward vaccination and reveal the reasons behind the poor vaccination uptake and resulting lower acceptance, which can inform the design of appropriate interventions [23]. Overall, for every ten participants in this multinational study, about three were found to feel confident about receiving the vaccine, nine showed no complacency toward the COVID-19 vaccine, five engaged in calculations, four demonstrated collective responsibility toward the COVID-19 vaccine, and 3.5% of them indicated constraints.

The 5C psychological antecedents were previously developed in German and English to measure the vaccines' psychological antecedents and determinants [23]. Then a protocol on culturally adapting and using it with other populations and groups was developed by the same group [36]. Ghazy et al. (2021) [35] then demonstrated that it had satisfactory discriminatory power to predict the psychological COVID-19 vaccine acceptance antecedents and identified a cutoff score.

This study found that the highest vaccine confidence was in the UAE, Saudi Arabia, and Kuwait, and the lowest confidence was in Egypt, possibly because a higher proportion of the populations in the UAE and Saudi Arabia had received at least one dose of the COVID-19 vaccination. However, even though the vaccination data is still unclear, it is reported that only around 6.5% of the population in Egypt has been fully vaccinated [38]. Five vaccines have been approved in the UAE; Sinopharm, Pfizer-BioNTech, Sputnik V, Oxford-AstraZeneca and Moderna [39]; and three have been approved in Saudi Arabia; Pfizer-BioNTech, Oxford-AstraZeneca, and Moderna [40]. However, five vaccines have been approved in Egypt; Sinopharm, Oxford-AstraZeneca, Johnson & Johnson, Sputnik V, and Sinovac [41]. As the population confidence discrepancy reflects the confidence in the COVID-19 vaccination and health authorities, the cultural and economic differences between the Gulf countries and other Arab countries may possibly affect the confidence of the population. Additionally, confidence is also affected by public education and awareness efforts that target precautions, focus on infection reduction, and stress the importance of vaccinations [42]. A recent study by Al-Sanafi & Sallam (2021) [43] found that health care workers in Kuwait had high intentions to take the vaccination, which may have been because of the high number of cases, vaccine availability, the public education efforts, and the policies imposed by the authorities. Elharake et al. [44] also found a high acceptance rate in Saudi Arabia for taking the vaccines in health care workers, with the male workers having a higher vaccination acceptance rate than the female workers, with reports of trusting the authorities as the main reason. Similar results for high HCP acceptance have also been found in Poland and Canada [45, 46]. However, a study on health care workers in Egypt found that only 3.5% were willing to take the vaccines and 40.9% claimed that they would not take the vaccine [47].

Complacency was higher in Morocco and Jordan, and lowest in Lebanon. Complacent people often believe that vaccination is not important as their immune system is able to protect them from being infected. It was found that because the Chinese believed that as they were currently healthy, they did not require the vaccination, which affected their intention to be

vaccinated [48]. The economic and political instability in Lebanon could also contribute to the way the people perceive the vaccines and their complacency.

The constraints were higher in Morocco and Iraq. It is possible that there are specific psychological barriers to taking the vaccine as Morocco has vaccinated nearly 46 million people, with 58.1% of the population being fully vaccinated. Morocco was also ranked the first in Africa to vaccinate its population as it received more vaccination doses than any other African country and established many vaccination centers and mobile vaccination teams [38, 49]. Since the beginning of the vaccination drive, Morocco's government has launched a communication campaign that provides relevant information and reassurance to encourage people to get vaccinated [49]. The main barrier to vaccination by the Moroccan people is the European refusal to acknowledge the Sinopharm and Sputnik V vaccines, which could restrict travel to Europe for people vaccinated with their doses [50].

The highest calculation was found in Sudan and Egypt, which refers to the weighing up of the benefits and risks of being vaccinated before making any decision. Sudan and Egypt have respectively fully vaccinated 1.3% and 8.1% people to date [38]. Amnesty International reported that there had been poor vaccine information provided in the local media and the Egyptian authorities, a limited awareness campaign, and a lack of a defined strategy and transparency for vaccine distribution, all of which could be potential reasons for the high calculation level [51].

UAE was found to have the highest collective responsibility toward the COVID-19 vaccine, that is, they were found to be more willing to protect themselves and others from being infected with the COVID-19 virus. The UAE has been promoting social responsibility and collective responsibility through educational and social programs in the past few years, its national charter also stresses the importance of social commitment and responsibility [52], and the UAE 2030 Sustainable Development Goals agenda incorporates goals to improve solidarity, unity, and social responsibility [53].

The predictors affecting these five psychological antecedents varied; however, they were mainly related to being male, being of advanced age, educated, being a HCP, having had COVID-19, or an infected relative or one that died from COVID-19. A preprint study in Arab countries found that females, middle-aged persons, lower education, and lack of knowledge regarding vaccine type had negative correlations with vaccine acceptance; while regularly receiving flu vaccines, working in health facilities, and high rates of COVID-19 infection had a positive correlation with vaccine acceptance [54]. Results from Turkey and the United Kingdom have found that being male, having a higher educational degree, and having children affected COVID-19 vaccine acceptance [55], and results from a low-and middle-income countries' study found that the willingness of people to be protected from COVID-19 was the main reason for vaccine acceptance, while the fear of side effects was the main reason for VH [32].

Sallam et al. (2021) [56] conducted a study in Jordan, Kuwait, and other Arab countries and found that the COVID-19 vaccine acceptance rate was only 29.4%, which was very low compared to 55% in Russia and the United States of America (USA) and 90% in China [57–59]. Khubchandani et al. [60] studied the COVID-19 VH in USA and found that 22% were hesitant, with these differences being related to gender, race, ethnicity, education, and social class. It was concluded that the high VH was due to belief that political and social factors and pressure were behind the accelerated approval of the COVID-19 vaccines before complete testing for their efficacy and safety. A systematic review found that there was variability in the COVID-19 vaccination acceptance between countries with many having an acceptance rate lower than 60%, which reflects the challenges ahead in controlling the COVID-19 pandemic. Lower rates were reported in the Middle East, Russia, and Eastern Europe, while higher rates were reported in East and Southeast Asia. It was concluded that COVID-19 VH has a major

role in controlling the pandemic, which in turn needs a collaborative response from governments, policymakers, and the media [61]. Another meta-analysis [62] found a VH of 17% and a pooled vaccine acceptance of 75%, and identified that the two reasons for vaccine acceptance were case fatalities and the number of COVID-19 cases, and the most powerful reason affecting the intention to be vaccinated was the people's trust in the safety of the vaccines provided in their country.

Soares et al. [63] studied the determinants of VH and found that young age, loss of income during the pandemic, no intention of taking influenza vaccines, low confidence in the health care system during the pandemic, the perception of the adequacy measures taken by the government, inadequate information given by health authorities, and a low confidence in the COVID-19 vaccine safety and efficacy were the main factors affecting the refusal or delay in taking the vaccine.

In the Arab world, a few studies have focused on COVID-19 VH. For example, in a recent pre-COVID-19 study conducted in the UAE, 12% of parents showed VH due to concerns related to the side effects, safety, and multiple injection sites [64]. In another qualitative study, HCPs in the UAE expressed VH and a need for training [65]. In Kuwait, many HCPs accepted the COVID-19 vaccine, with the VH concentrated mainly in female HCPs, nurses, and those working in private facilities [43]. In Egypt, the level of VH in medical students was reported at 46.0% with the main concerns being side effects and the ineffectiveness of COVID-19 vaccine [66]. In Jordan, El-Elimat et al. [67] conducted an online survey on the willingness to take the COVID-19 vaccine, and found that 37.4% were willing, 36.3% were unwilling, and 26.3% were indecisive, with many having a greater vaccine acceptance for the elderly than for themselves.

While vaccine development and availability are necessary to achieve immunity against a disease such as COVID-19, they are not sufficient. Therefore, reducing the incidence and prevalence of COVID-19 requires high vaccine acceptance and coverage to ensure high population acceptance [68–71].

## Strength and limitations

One of the major strengths of the present study was the large sample size and the diversity of the survey population in terms of the range of countries, age groups, and ethnic and cultural backgrounds. While a validated Arabic version of the 5C scale was employed to guarantee the internal consistency of the study results, the use of convenience sampling and the online distribution of the study tool limited the generalization of the study results to the region of interest. Further, there was a risk of selection bias as it favored only those who had access to the internet, and because a self-reported questionnaire was used to collect the data, the findings may have been affected by a social desirability bias. Despite the stated limitations, the study findings were consistent with previous studies that have reported the behavioral factors associated with COVID-19 VH. More importantly, this study was able to shed light on the overlooked psychological antecedents of COVID-19 vaccination behavior in Arab countries, which can guide the development of better policies.

## Conclusions

This study found wide variations in the psychological antecedents of COVID-19 vaccination between the studied Arab countries. The vaccine confidence and collective responsibility were higher in the countries that had high vaccination rates and lower in the countries that had low vaccination rates. However, the other psychological parameters (complacency, constraints, and calculation) differed across the countries that had varying vaccination rates. Gender, education, being infected, or having had an infected relative or one that died because of COVID-

19 were the predictors affecting the five psychological vaccination antecedents. Therefore, government decisions and policies, the media, and health care authorities must play a role in changing the attitude of the population toward COVID-19 vaccines to ensure optimal vaccine acceptance.

## Supporting information

**S1 Table. Multivariate analysis of predictors affecting the 5C psychological antecedents.** (DOCX)

## Acknowledgments

The authors would like to acknowledge and thank all study participants. The authors also would like to thank Prof. Dr. Nessrin El-Nimr for her valuable opinions that were added to this manuscript. The authors would like to thank the following contributors: Dr. Samar Samy Abd ElHafeez (Egypt), Dr. Ramy Shaaban (Egypt), Dr. Ahmed Nour Eldin (Egypt), Dr. Mohamed Abdelbaky (Egypt), Dr. Omar Elenezy (Kuwait), Dr. Emad Shaqoora (Palestine), Dr. Haydar Elhamzawi (Iraq), Dr. Majed Alharthi (Saudi Arabia), Dr. Mohamed Yacoub (Egypt), Dr. Rony Ibrahim (Egypt), Dr. Ismail Ibrahim (Kuwait), Dr. Mohamed Alamin (Mauritania) and Dr. Alaa Hamdy (Egypt).

## Author Contributions

**Conceptualization:** Marwa Shawky Abdou, Ramy Mohamed Ghazy.

**Data curation:** Marwa Shawky Abdou, Khalid A. Kheirallah, Maged Ossama Aly, Yasir Ahmed Mohammed Elhadi, Iffat Elbarazi, Ehsan Akram Deghidy, Haider M. El Saeh, Karem Mohamed Salem, Ramy Mohamed Ghazy.

**Formal analysis:** Ahmed Ramadan.

**Methodology:** Marwa Shawky Abdou, Ramy Mohamed Ghazy.

**Software:** Ahmed Ramadan.

**Supervision:** Marwa Shawky Abdou.

**Visualization:** Ahmed Ramadan.

**Writing – original draft:** Marwa Shawky Abdou, Khalid A. Kheirallah, Maged Ossama Aly, Karem Mohamed Salem, Ramy Mohamed Ghazy.

**Writing – review & editing:** Marwa Shawky Abdou, Khalid A. Kheirallah, Yasir Ahmed Mohammed Elhadi, Iffat Elbarazi, Ehsan Akram Deghidy, Haider M. El Saeh, Ramy Mohamed Ghazy.

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
