## [Decision Letter · Decision Letter 0]

20 Sep 2021

PONE-D-21-26350Psychological antecedents towards COVID-19 vaccination using the Arabic 5C validated tool: An online study in 13 Arab countriesPLOS ONE

Dear Dr. Abdou,

Thank you for submitting your manuscript to PLOS ONE. After careful consideration, we feel that it has merit but does not fully meet PLOS ONE’s publication criteria as it currently stands. Therefore, we invite you to submit a revised version of the manuscript that addresses the points raised during the review process.

We look forward to receiving your revised manuscript.

Kind regards,

Sanjay Kumar Singh Patel, Ph.D.

Academic Editor

PLOS ONE

Journal Requirements:

Reviewers' comments:

Reviewer's Responses to Questions

Reviewer #1: The manuscript by Abdou et al. “Psychological antecedents towards COVID-19 vaccination using the Arabic 5C validated tool: An online study in 13 Arab countries” is noteworthy and the manuscript requires minor revision before its publication.

Comments

1. The English of manuscript can be polished (minor).

2. Please avoid the uses of standard deviation in the text.

3. Page 4 (first paragraph), COVID-19 basic information’s may be provided with citations like symptoms, mortality and preventions approaches i.e. distancing and improving health, and future challenges by their high mutation ability and advanced treatments (doi: 10.1371/journal.pone.0231236; doi: 10.1007/s12088-020-00893-4; doi: 10.1007/s12088-020-00908-0).

4. Page 5 (first paragraph), authors may provide details various vaccines COVID-19 prevention’s efficiency and ahead challenges. Also, use this information for discussion (minor).

5. The objectives of this study are not clear.

6. Fig. 2 quality may be improved (high resolution).

Reviewer #2: The authors reports a study titled “Psychological antecedents towards COVID-19 vaccination using the Arabic 5C validated tool: An online study in 13 Arab countries”. In this report the authors used social media platforms such as Facebook, Twitter, and WhatsApp to collect data for this study. More than 4000 participants from 13 Arabic countries took part in this survey.

Minor comments:

1. The authors states that “In this study, it was found that the highest confidence was among the population from UAE, Saudi Arabia, and Kuwait, while the lowest confidence was among the population from Egypt.” The authors further note that less than 4% of Egypt’s population had been vaccinated as compared to UAE and Saudi Arabia where majority of the populations were vaccinated. The authors should clarify in the manuscript, that it’s possible that the low confidence among the population from Egypt was due to unavailability of vaccines and information.

2. The entire study is based on data obtained from online platforms. However, it’s unclear to me, the % of population using these services in low-income countries. Therefore, this study cannot be generalized.

---

## [Author Response · Author response to Decision Letter 0]

30 Oct 2021

Dear Prof Emily Chenette

Editor in-Chief of Plos One,

Thank you for giving me the opportunity to submit a revised draft of my manuscript titled [The Coronavirus Disease 2019 (COVID-19) vaccination psychological antecedent assessment using the Arabic 5c validated tool: An online study in 13 Arab countries] to [Plos One]. We appreciate the time and effort that you and the reviewers have dedicated to providing your valuable feedback on our manuscript. We are grateful to the reviewers for their insightful comments on our paper. We have been able to incorporate changes to reflect most of the suggestions provided by the reviewers. We have highlighted the changes within the manuscript. Here is a point-by-point response to the reviewers’ comments and concerns.

This is our response to the comments raised by the editors and reviewers

Comment: 1.1 Please ensure that your manuscript meets PLOS ONE's style requirements, including those for file naming. 

Response: Dear Editor, Thank you for this comment. I revised the manuscript to guarantee that it fulfills the PLOS-ONE journal style.

 Comment 1.2: Please provide additional details regarding participant consent. In the ethics statement in the Methods and online submission information, please ensure that you have specified what type you obtained (for instance, written or verbal, and if verbal, how it was documented and witnessed). If your study included minors, state whether you obtained consent from parents or guardians. If the need for consent was waived by the ethics committee, please include this information.

Response: in the methodology section, we added the following sentence “All participants were informed that their participation was voluntary, and consent was obtained by answering a question prior to administering the survey.” Page 9 line 185-186

Comment 1.3: Please review your reference list to ensure that it is complete and correct. If you have cited papers that have been retracted, please include the rationale for doing so in the manuscript text, or remove these references and replace them with relevant current references. Any changes to the reference list should be mentioned in the rebuttal letter that accompanies your revised manuscript. If you need to cite a retracted article, indicate the article’s retracted status in the References list and also include a citation and full reference for the retraction notice.

Response1.3: thank you for this notice. Indeed, we did not cite any retracted reference in the manuscript, however, we updated some reference to reflect recent data of COVID 19 incidence, mortality, and vaccination.

Second: Response to reviewers' comments:

Reviewer #1: 

Comment 2.1 The English of manuscript can be polished (minor).

Response 2.1 Thank you very much for your suggestion, we agree with this comment therefore, we revised the abstract and modified it as possible to suitable for publication in your estimated journal. We have sent the manuscript to the Egyptian knowledge bank for lingual editing. A service provided by native speaker through springer.

Comment 2.2: Please avoid the uses of standard deviation in the text.

Response 2.2: Done

Comment 2.3: Page 4 (first paragraph), COVID-19 basic information’s may be provided with citations like symptoms, mortality and preventions approaches i.e. distancing and improving health, and future challenges by their high mutation ability and advanced treatments (doi: 10.1371/journal.pone.0231236; doi: 10.1007/s12088-020-00893-4; doi: 10.1007/s12088-020-00908-0).

Response 2.3: : I would like to thank you for this important observation. We used the following reference (Deploying biomolecules as anti-COVID-19 agents), however, we think the other two references may be irrelevant (Diet, Gut Microbiota and COVID-19 & Forecasting the novel coronavirus COVID-19).

Comment 2.4: Page 5 (first paragraph), authors may provide details various vaccines COVID-19 prevention’s efficiency and ahead challenges. Also, use this information for discussion (minor).

Response 2.3: Done Page 4 lines 83-91.

2.5 The objectives of this study are not clear.

Response 2.5: Thank you for your comment. We have added the following paragraph “At this stage in the pandemic, especially as vaccine compliance remains variable and inconsistent, public health officers and policymakers, especially in developing countries where healthcare resources are limited, need to understand the reasons and factors associated with VH. This study was therefore developed to investigate the psychological antecedent factors in Arab populations toward the COVID-19 vaccination” page 6 line 120-124.

Comment 2.6 Fig. 2 quality may be improved (high resolution).

Response 2.6: Dear reviewer, we have uploaded the same figure with higher resolution based on your recommendation.

Reviewer #2: 

Comment 3.1: The authors states that “In this study, it was found that the highest confidence was among the population from UAE, Saudi Arabia, and Kuwait, while the lowest confidence was among the population from Egypt.” The authors further note that less than 4% of Egypt’s population had been vaccinated as compared to UAE and Saudi Arabia where majority of the populations were vaccinated. The authors should clarify in the manuscript, that it’s possible that the low confidence among the population from Egypt was due to unavailability of vaccines and information.

Response 3.1 Thank you for your comment. We had the following paragraph written and was modified based on your suggestions.” Amnesty International reported that there had been poor vaccine information provided in the local media and the Egyptian authorities, a limited awareness campaign, and a lack of a defined strategy and transparency for vaccine distribution, all of which could be potential reasons for the high calculation level [51].” page 25 line 336-339.

Comment 3.2: The entire study is based on data obtained from online platforms. However, it’s unclear to me, the % of population using these services in low-income countries. Therefore, this study cannot be generalized.

Response 3.2: we totally agree with you and we have mentioned this in the limitation of the study.” Further, there was a risk of selection bias as it favored only those who had access to the internet, and because a self-reported questionnaire was used to collect the data, the findings may have been affected by a social desirability bias” page 30 line 381-383. However, the proportion of Arab population who have access to internet ranges from 50.5% in Libya To 95.7% in Saudi Arabia (Data reportal, 2021; https://datareportal.com/reports/?tag=Local).

---

## [Editor Report · Decision Letter 1]

8 Nov 2021

The Coronavirus Disease 2019 (COVID-19) vaccination psychological antecedent assessment using the Arabic 5c validated tool: An online survey in 13 Arab countries

PONE-D-21-26350R1

Dear Dr. Abdou,

We’re pleased to inform you that your manuscript has been judged scientifically suitable for publication and will be formally accepted for publication once it meets all outstanding technical requirements.

Kind regards,

Sanjay Kumar Singh Patel, Ph.D.

Academic Editor

PLOS ONE

---

## [Editor Report · Acceptance letter]

15 Nov 2021

PONE-D-21-26350R1 

The Coronavirus Disease 2019 (COVID-19) vaccination psychological antecedent assessment using the Arabic 5c validated tool: An online survey in 13 Arab countries 

Dear Dr. Abdou:

I'm pleased to inform you that your manuscript has been deemed suitable for publication in PLOS ONE. Congratulations! Your manuscript is now with our production department. 

Kind regards, 

on behalf of

Dr. Sanjay Kumar Singh Patel 

Academic Editor

PLOS ONE